# SKFlow: Learning Optical Flow with Super Kernels

**Shangkun Sun** [1]    **Yuanqi Chen** [1]    **Yu Zhu** [2]    **Guodong Guo** [2,3]    **Ge Li** [1]

[1]School of Electronic and Computer Engineering, Peking University
[2]Institute of Deep Learning, Baidu Research [3]West Virginia University

## Abstract

Optical flow estimation is a classical yet challenging task in computer vision. One of the essential factors in accurately predicting optical flow is to alleviate occlusions between frames. However, it is still a thorny problem for current top-performing optical flow estimation methods due to insufficient local evidence to model occluded areas. In this paper, we propose the Super Kernel Flow Network (SKFlow), a CNN architecture to ameliorate the impacts of occlusions on optical flow estimation. SKFlow benefits from the super kernels which bring enlarged receptive fields to complement the absent matching information and recover the occluded motions. We present efficient super kernel designs by utilizing conical connections and hybrid depth-wise convolutions. Extensive experiments demonstrate the effectiveness of SKFlow on multiple benchmarks, especially in the occluded areas. Without pre-trained backbones on ImageNet and with a modest increase in computation, SKFlow achieves compelling performance and ranks **1st** among currently published methods on the Sintel benchmark. On the challenging Sintel clean and final passes (test), SKFlow surpasses the best-published result in the unmatched areas (7.96 and 12.50) by $9.09\%$ and $7.92\%$. The code is available at https://github.com/littlespray/SKFlow.

## 1   Introduction

Optical flow is the task of modeling per-pixel motion across a pair of frames with various downstream applications, e.g., pedestrian re-identification, video segmentation, scene reconstruction, etc. Nowadays, optical flow estimation still faces tough challenges such as occlusions and motion blurs, etc. Among those challenges, occlusion is considered one of the most difficult problems which are still under-explored. Notably, the term *occlusion* in optical flow estimation can be extended to a broader definition [16]: *regions where pixels appear in the current frame while disappearing in the next frame.* Occlusions pose apparent difficulties in predicting optical flow since it directly violates the brightness constancy constraint [10], where the intensities of pixels are regarded as the same between consecutive frames. Therefore, the correspondence matching of occluded areas between frames could be extremely hard.

One of the potential solutions is to utilize the neighboring pixels to recover motions of the occluded regions, such as learning the neighboring relationship by CNNs  [31, 32, 36], or interpolating the hidden motions via smoothness terms in Markov Random Fields [4]. However, both convolution and interpolation are restricted to the local information within the small operation window, which only focuses on learning the local evidence. As occlusions get worse, the local evidence would be insufficient to recover the hidden motion and thus severely degrade the performance. Recent works [16, 42] propose to model long-range dependencies between local descriptors via non-local methods to make up for the missing local evidence. These methods alleviate the issue to some extent but still tend to fail since the representative capabilities of local descriptors have been largely weakened when facing severe occlusions.

36th Conference on Neural Information Processing Systems (NeurIPS 2022).

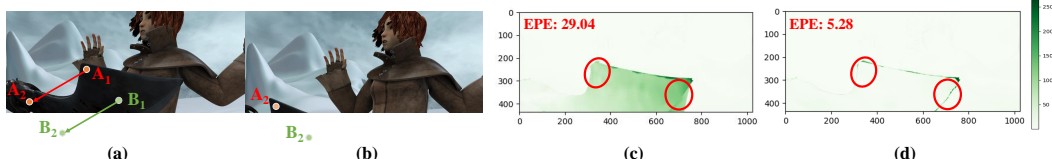

Figure 1: Occlusions between frames and flow error map of different methods. From (a) to (b), the most of the blade moves out-of-frame (e.g. pixel $B_1$ moves to pixel $B_2$). But the motion of the occluded $B_1$ could still be recovered via the non-occluded motion of $A_1$ over long distances due to the large receptive fields. (c) and (d) denote the error between the predicted flow and ground truth from method with normal convolutions (e.g. GMA) and our SKFlow, respectively. The darker the color, the greater the error. SKFlow achieves lower end-point error (EPE) and performs better in the occluded blade area.

Motivated by [21, 7] that larger kernels present an effective way compared to deeper layers for larger receptive fields, using large kernels in an optical flow estimation network is arguably a possible solution to handle the occlusion problem, as illustrated in Figure 1. However, directly applying large kernels is not practical due to: **(1)** The quadratic growth in computation. As the kernel size grows from $3 \times 3$ to $15 \times 15$, the model would expand 25 times its original size. **(2)** The optimization issue. As shown in [7, 24, 11], even with careful design and training on a huge dataset, large kernel networks take effort to optimize and are even prone to a performance drop. It is more challenging on optical flow networks, where the training data is often much smaller. For example, Sintel [3] dataset has only $1,041$ training samples.

In this work, we propose Super Kernel Flow Network (SKFlow), where we introduce a new architecture design that efficiently utilizes the conical connections and hybrid depth-wise convolutions, and accordingly develop an effective optical flow network to handle the occlusions. A detailed description of the approach is described in Section 3. SKFlow is better at resolving the ambiguity caused by severe occlusions and obtains compelling performance on standard benchmarks (see Section 4). Besides, SKFlow attains a good trade-off between accuracy and computation cost. On the challenging Sintel final pass test set, SKFlow ranks **1st** among all published methods at the time of submission, with the increase in MACs less than **8.42%** compared with GMA [16].

Our contributions are summarized as follows: **(1)** We introduce the super kernel schemes to the optical flow task for the first time. **(2)** We explore three new architecture designs for super kernel designs in the optical flow network and proposed a new network which we named SKFlow. **(3)** Our proposed SKFlow achieves state-of-the-art performance on standard benchmarks and obtains strong cross-dataset generalization.

## 2 Related work

**Optical flow estimation.** Traditionally, optical flow is formulated as an energy minimization problem. Based on brightness constancy and spatial smoothness, Horn and Schunck [10] pioneered the variational approach to computing optical flow. On that basis, there emerged subsequent improvements for visual similarity designs and regularization terms [10, 1, 2, 30]. In the deep learning era, CNNs have emerged as a powerful technique for optical flow estimation since FlowNet [8]. Then the coarse-to-fine strategy is widely adopted [31, 32, 12, 13, 40, 14, 43]. Coarse-to-fine methods are influential but tend to miss small motions due to the inaccurate flow guidance at the coarse level. Therefore, RAFT [36] presents an iterative refinement method with the all-pairs field transform. It maintains a single fixed flow field at high resolutions and achieves significant improvements, inspiring lots of works such as Flow1D [39], FM-RAFT [17], Separable flow [42], AGFlow [20], and GMA [16], etc.

**Kernel sizes in convolution layers.** Recently, there have emerged some explorations of applying large kernels to CNNs, which have not been widely used since early designs [34, 35, 33, 27]. The ERF theory [21] indicates that larger kernels are more effective to obtain larger receptive fields than deeper layers. LRNet [11] proposes the local relation layer where $7 \times 7$ convolutions are widely used. However, there is a performance drop when kernel size is increased to $9 \times 9$. GCN [24]

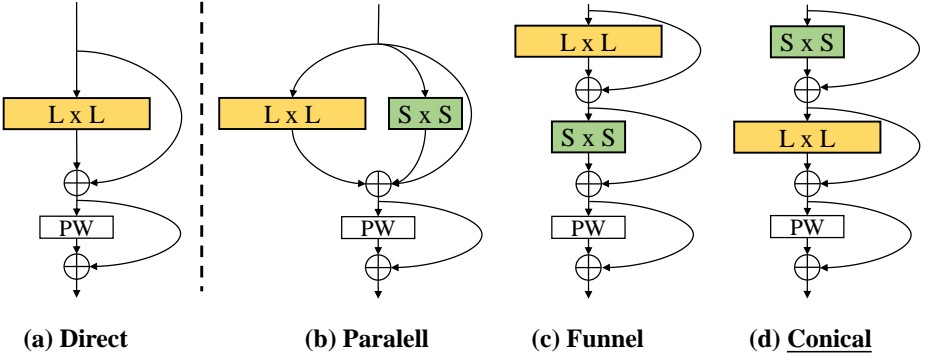

| **(a) Direct** | **(b) Paralell** | **(c) Funnel** | **(d) Conical** |

Figure 2: Architecture designs for super kernel blocks. $L$ and $S$ denote the kernel size of large and the auxiliary depth-wise convolutions respectively. PW is the point-wise convolution. The underlined design is used in our SKFlow model.

further enlarges the kernel size in the segmentation task by dense-connected symmetric separable convolutions. It is observed that the performance tends to decrease when transferred to other tasks. CKCNN [26] and FlexConv [25] adopt larger kernel sizes, while the model size and computation cost grow correspondingly. RepLKNet [7] develops the re-parameterization technique to help optimize large kernels and scales the kernel size to $31 \times 31$, achieving state-of-the-art performance on classification and downstream tasks such as detection and segmentation. Despite its competitive results on these high-level vision tasks, our experiments show that it could not be directly applied for dense prediction tasks such as optical flow estimation (see Section 4.4). Therefore, we explore the large kernel designs in the optical flow network, aiming at alleviating the occlusions and improving the estimation performance.

## 3 Approach

We propose to utilize the large receptive field to handle the local ambiguity caused by occlusions. Since the motion of the occluded pixels is hard to estimate with local ambiguity, our SKFlow method introduces super kernels to help complement local evidence, and an efficient architecture design is proposed to reduce the computation burden. We describe the details of the super kernel block in Section 3.1. The overall network architecture is then presented in Section 3.2. Finally, the learning process of SKFlow is demonstrated in Section 3.3.

### 3.1 Super kernel block designs

We enlarge the kernel size instead of deepening layers to attain a large receptive field given the following considerations. **(1)** Deep layers lead to the optimization issue. Although ResNet [9] has resolved the optimization problems for deeper networks [7, 9] to some extent, recent works [6, 37] still suggest that it might behave like shallow networks and obtain the limited receptive fields with deeper layers. **(2)** The Effective Receptive Field (ERF) [21] grows sub-linearly with the number of layers and linearly with the kernel size, which demonstrates that the large kernel size is more effective.

**Super kernel block components.** In order to reduce the massive computation cost caused by large kernel sizes, we propose the super kernel block that contains three components: **(1)** Hybrid depth-wise convolution kernels. Inspired by separable convolutions [5], we first split a convolution into a couple of depth-wise convolutions. Namely, a large depth-wise kernel with size $L \times L$ and an auxiliary small depth-wise kernel with size $S \times S$. With the input of shape $N \times C_{in} \times H \times W$, the computation cost for this hybrid depth-wise convolutions is $N \times C_{in} \times H \times W \times (L^2 + S^2) \times 1$. The auxiliary kernel is designed to help capture small-scale patterns in the frame. **(2)** Residual connections. Residuals [9] are used (a) to combine the large kernel and the auxiliary kernel; (b) connect depth-wise and point-wise convolutions. **(3)** Point-wise convolution. An $1 \times 1$ point-wise convolution is applied after hybrid depth-wise convolutions to help the information flow across channels. The point-wise convolution

does not change the input dimension and the computation cost is: $N \times C_{in} \times H \times W \times C_{in}$ with the input shape $N \times C_{in} \times H \times W$.

**Super kernel block architecture.** We propose three architecture designs for our super kernel blocks. As shown in Figure 2, our designs ($b$, $c$, and $d$ columns) are derived from the Direct design in column $a$. Specifically, **(1) Parallel** (Figure 2 (b)), is a block of layers that adopting a parallel small kernel in the large depth-wise convolution layer. A point-wise convolution is followed by parallel VGG-style convolutions combined with skip connections. **(2) Funnel** (Figure 2 (c)), is a ResNet-style block of layers where the kernel sizes are decreased from large kernel $L$ to auxiliary kernel $S$. **(3) Conical** (Figure 2 (c)), is similar to the Funnel design but applies the hybrid convolutions in the opposite order. Experimental results in Section 4.4 that all three large kernel designs outperform normal convolution layers with normal small kernels and the computation increase is very limited. Among all these architectures, the conical block obtains the best performance and is adopted as our final design, which can be formulated as:

$$\mathbf{h} = \sigma(\mathbf{x} + Conv_{S \times S}^{dw}(\mathbf{x})) \tag{1}$$

$$\mathbf{d} = \sigma(\mathbf{h} + Conv_{L \times L}^{dw}(\mathbf{h})) \tag{2}$$

$$\mathbf{p} = Conv_{1 \times 1}^{pw}(\mathbf{d}) \tag{3}$$

$$\mathbf{o} = \mathbf{d} + \sigma(\mathbf{p}) \tag{4}$$

where $\mathbf{x}$ and $\mathbf{o}$ denote the input and output feature map, respectively. $Conv_{S \times S}^{dw}$, $Conv_{L \times L}^{dw}$ refer to the depth-wise convolution with large and small kernels, respectively. $Conv_{1 \times 1}^{pw}$ is the point-wise convolution and $\sigma$ is the activation function.

To match the dimension and increase more non-linear transforms, two pairs of $1 \times 1$ convolutions are performed in each super kernel block. Each pair of convolution layers will first transform the input feature map from dimension $C_{in}$ to $\alpha C_{in}$, and then from $\alpha C_{in}$ to $C'$. Namely, the computation cost is $N \times H \times W \times C_{in} \times \alpha C_{in} + N \times H \times W \times \alpha C_{in} \times C'$, where $C' = C_{in}$ for the first pair of convolution layers and $C' = C_{out}$ for the second pair. Therefore, the computation is $N \times H \times W \times \alpha C_{in} \times (C_{in} + C_{out} + C_{in} + C_{in})$. In summary, the total computation of a block is $N \times H \times W \times C_{in} \times [L^2 + S^2 + C_{in} + \alpha(3C_{in} + C_{out})]$. We can then compute the computation cost ratio of our super kernel block and the normal convolution as:

$$\begin{aligned} \frac{Cost_{ours}}{Cost_{normal}} &= \frac{N \times H \times W \times C_{in} \times [L^2 + S^2 + C_{in} + \alpha(3C_{in} + C_{out})]}{N \times H \times W \times C_{in} \times C_{out} \times L^2} \\ &= \frac{1}{C_{out}} + \frac{1}{C_{out}}\frac{S^2}{L^2} + \frac{1 + 3\alpha}{L^2}\frac{C_{in}}{C_{out}} + \frac{\alpha}{L^2} \end{aligned} \tag{5}$$

where $\alpha$, $C_{in}$ and $C_{out}$ are constants, and $C_{out} \gg 1$ in practice. Notably, the computation cost of our super kernel is reduced to $O(1/L^2)$ compared with normal large kernels. Besides, our super kernel also achieves compelling performance (see Section 4 for results).

## 3.2 Super kernel flow network

Our Super Kernel Flow (SKFlow) network follows a similar framework to GMA [16] except the super kernel modules, which are made up of super kernel blocks. We present the overall network architecture in the following.

**All-pairs correlation cost volume.** The all-pairs correlation cost volume proposed by [36] is used to model correlations for all possible displacements. Features from two frames perform a dot-product and then the matching correlations $\mathbf{c}$ across different levels can be built, namely,

$$\mathbf{c}^l(i, j, m, n) = \frac{1}{2^{2l}} \sum_{u}^{2^l} \sum_{v}^{2^l} \left\langle \mathbf{x}_1(i, j), \mathbf{x}_2(2^l m + u, 2^l n + v) \right\rangle \tag{6}$$

where $\mathbf{x}_1$ and $\mathbf{x}_2$ are feature maps extracted from input frames, and $l$ refers to the correlation level. $\langle, \rangle$ denotes the inner product function.

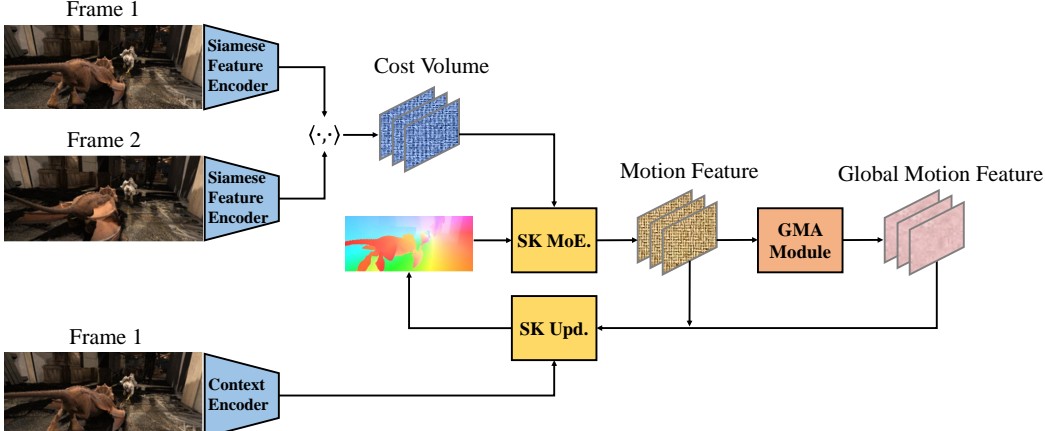

Figure 3: Overview of our proposed SKFlow. SK MoE. and SK Upd. denote our proposed super kernel motion encoder and updater respectively. GMA module is the non-local method proposed in [16].

**Global motion aggregation (GMA) module.** The GMA module adopts the self-attention mechanism to model long-range connections between local descriptors. Following GMA [16], we apply the module in the flow decoder. Given the input motion feature $\mathbf{x}$ with height $H$ and width $W$, the output $\mathbf{o}$ is defined as:

$$\mathbf{o}(i,j) = \mathbf{x}(i,j) + \gamma \sum_{u}^{H} \sum_{v}^{W} f(\mathbf{q_y}(i,j), \mathbf{k}_y(u,v)) \mathbf{v_x}(u,v) \tag{7}$$

where $\mathbf{q_y}$ and $\mathbf{k}_y$ denote the query and key vector derived from the context feature map. $V_m$ is the value vector derived from $\mathbf{x}$, $f$ is the dot-product attention function and $\gamma$ is a learnable coefficient.

**Super kernel modules.** Super kernel modules consist of **(1) Super kernel motion encoder**, which shares similar architectures with GMA but applies super kernel blocks. It derives motion features from cost volume vectors $\mathbf{c}$ and the predicted flow $\mathbf{f}$, namely,

$$\mathbf{c}' = SKBlock(SKBlock(\mathbf{c})) \tag{8}$$

$$\mathbf{f}' = SKBlock(SKBlock(\mathbf{f})) \tag{9}$$

$$\mathbf{o} = Concat(SKBlock(Concat(\mathbf{c}', \mathbf{f}')), \mathbf{f}) \tag{10}$$

where $SKBlock$ denotes our proposed super kernel block, and $Concat$ is the concatenation operation. **(2) Super kernel updater.** Different from the ConvGRU used in RAFT and GMA, we directly adopt our super kernel block as the updater to refine the predicted residual flow $\Delta\mathbf{f}$, which could be formulated as:

$$\Delta\mathbf{f} = SKBlock(Concat(\mathbf{x}_m, \mathbf{x}_c, \mathbf{x}_g)) \tag{11}$$

where $\mathbf{x}_m$, $\mathbf{x}_c$ denote the motion feature and context feature. $\mathbf{x}_g$ refers to the global motion feature from the GMA module.

The whole architecture of our SKFlow is shown in Figure 3. It is notable that the super kernel blocks are applied in the decoder only based on the following consideration: Applying in the decoder is more efficient due to its relatively small size. Given the lack of efficient implementation like a normal $3 \times 3$ kernel for large depth-wise kernels in most libraries, super kernels in those two large encoders can take lots of additional time in inference and training. Although it may be solved when faster implementations occur in the future, currently applying super kernels in the smaller decoder part brings a modest time increase and achieves significant performance improvements.

### 3.3 Supervision

Following previous works [36, 16], we adopt the following loss function to supervise the parameter update. The $l_1$ loss of the predicted flow after every refinement is weighted with the exponentially

Table 1: Quantitative results on Sintel and KITTI. We report average End-Point Error(EPE) unless specified. C+T refers to the FlyingChairs → FlyingThings schedule. + S + K + H denotes that Sintel, KITTI, and HD1K training sets are combined when finetuning on Sintel. $^\star$ denotes using warm-start strategy in RAFT [36]. SKFlow achieves the best generalization performance on C+T and C+T+S+K+H.

| Training Data | Method | Sintel (train) | | KITTI-15 (train) | | Sintel( test) | | KITTI-15 (test) |
|---|---|---|---|---|---|---|---|---|
| | | Clean | Final | Fl-epe | Fl-all | Clean | Final | Fl-all |
| C+T | HD3 [41] | 3.84 | 8.77 | 13.17 | 24.0 | - | - | - |
| | PWC-Net [31] | 2.55 | 3.93 | 10.35 | 33.7 | - | - | - |
| | VCN [40] | 2.21 | 3.68 | 8.36 | 25.1 | - | - | - |
| | MaskFlowNet [43] | 2.25 | 3.61 | - | 23.1 | - | - | - |
| | FlowNet2 [15] | 2.02 | 3.54 | 10.08 | 30.0 | 3.96 | 6.02 | - |
| | DICL-Flow [38] | 1.94 | 3.77 | 8.70 | 23.6 | - | - | - |
| | RAFT [36] | 1.43 | 2.71 | 5.04 | 17.4 | - | - | - |
| | AGFlow [20] | 1.31 | 2.69 | 4.82 | 17.0 | - | - | - |
| | Separable Flow [42] | 1.30 | 2.59 | 4.60 | 15.9 | - | - | - |
| | GMA [16] | 1.30 | 2.74 | 4.69 | 17.1 | - | - | - |
| | **SKFlow (Ours)** | **1.22** | **2.46** | **4.27** | **15.5** | - | - | - |
| C+T+S+K+H | LiteFlowNet2 [13] | (1.30) | (1.62) | (1.47) | (4.8) | 3.48 | 4.69 | 7.74 |
| | PWC-Net+ [32] | (1.71) | (2.34) | (1.50) | (5.3) | 3.45 | 4.60 | 7.72 |
| | MaskFlowNet [43] | - | - | - | - | 2.52 | 4.17 | 6.10 |
| | RAFT [36] | (0.76) | (1.22) | (0.63) | (1.5) | 1.61$^\star$ | 2.86$^\star$ | 5.10 |
| | AGFlow [20] | (0.65) | (1.07) | (0.58) | (1.2) | 1.43 | 2.47 | 4.89 |
| | Separable Flow [42] | (0.69) | (1.10) | (0.69) | (1.6) | 1.50 | 2.67 | **4.64** |
| | GMA [16] | (0.62) | (1.06) | (0.57) | (1.2) | 1.39$^\star$ | 2.47$^\star$ | 5.15 |
| | **SKFlow (Ours)** | **(0.52)** | **(0.78)** | **(0.51)** | **(0.94)** | **1.28$^\star$** | **2.27$^\star$** | 4.84 |

increasing coefficients:

$$\mathcal{L} = \sum_{i=1}^{N} \lambda^{N-i} \, ||f_i - f_{gt}||_1 \tag{12}$$

where $|| \cdot ||_1$ denotes the $l_1$ distance between flow ground-truth $f_{gt}$ and output flow, and $f_i$ represents the predicted flow field at the $i$th refinement. $\lambda$ denotes weights on different predicted flows and is set to 0.8 in our experiments.

## 4    Experiments

**Experimental setup.**    Our model is evaluated on Sintel [3] and KITTI [22] datasets. We adopt the average End-Point-Error (EPE) as the evaluation metric, which denotes the mean flow error over all pixels. KITTI also adopts the Fl-All (%) metric, which computes the percentage of pixels with EPE larger than 3 pixels or over 5% of ground truth. Sintel consists of different passes rendered with different levels of difficulty. Specifically, the Albedo pass is rendered without illumination effects and has approximately piecewise constant colors. The clean pass introduces reflection properties and various illumination. The final pass adds motion blur, atmospheric effects, and other artistic embellishments to the lighting, which is the most challenging. We use the clean and final passes for training and evaluate our methods on all three passes. Notably, our SKFlow design scheme works both on GMA and RAFT networks. Detailed discussion on the RAFT-based SKFlow method is shown in the supplementary materials.

**Implementation details.**    Following previous works [36, 16, 42, 20], we first pre-train our SKFlow using the FlyingChairs → FlyingThings schedule, and then fine-tune for Sintel with the combined dataset from Sintel, FlyingThings, KITTI and HD1K [18]. Finally, we finetune our model on KITTI. We adopt the AdamW [19] optimizer and one-cycle policy [28]. Our SKFlow is built with PyTorch [23] library and trained using two Tesla V100 GPUs.

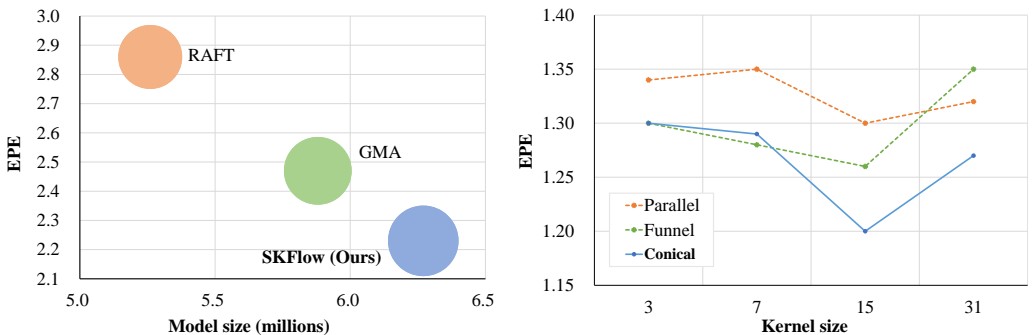

Figure 4: (a) Comparisons between MACs, parameters, and performance. A larger bubble represents higher MACs. (b) The comparison between the three designs with different kernel sizes. The Conical style achieves the lowest error with a $15 \times 15$ kernel size.

## 4.1 Quantitative results

Following previous works [36, 16, 42, 20], we evaluate our SKFlow on Sintel and KITTI-2015 dataset. We first evaluate the generalization ability of models by pretraining on FlyingChairs and FlyingThings datasets and evaluate on Sintel and KITTI datasets. Next, we compare the performance of each dataset. Models are finetuned on the training set combined from Sintel, HD1K, and KITTI datasets, and evaluated on the Sintel and KITTI datasets. The experimental results are shown in Table 1.

**Results for cross-dataset generalization.** Our SKFlow is first pre-trained on FlyingChairs and FlyingThings datasets and then evaluate on the Sintel and KITTI-2015 datasets. The cross-dataset results can be seen in Tables 1 "C+T" rows. From the results, we can see the generalization capability of different methods. Specifically, on the challenging Sintel final pass, our SKFlow obtains the best performance among all published methods, outperforming GMA by $10.21\%$. On KITTI and clean pass, our method achieves the Fl-epe of 4.27 and Fl-all of 15.5, and the EPE of 1.22, outperforming GMA by $8.96\%$, $9.36\%$ and $6.15\%$, respectively.

**Results on Sintel benchmark.** The experimental results are shown in Table 1 "C+T+S+K+H" rows. From the results, we can see that our SKFlow achieves significant improvements in performance on both Sintel training and test sets among the listed methods. Compared with GMA, our SKFlow achieves the EPE reduction of $7.91\%$ and $8.10\%$ on the clean and final pass of the Sintel test set. On the training clean and final pass, SKFlow outperforms GMA by $16.13\%$ and $26.42\%$.

**Results on KITTI benchmark.** The results are shown in Table 1 "C+T+S+K+H" rows. Our SKFlow achieves $4.84$, achieving competitive results. On the KITTI training set, our method achieves the improvement on GMA with the EPE of $0.51\%$ and of Fl-epe of $0.94\%$. On the KITTI test set, our method outperforms GMA by $6.02\%$.

## 4.2 Model efficiency

We then compare the efficiency of our SKFlow with current state-of-the-art methods. We use MACs, runtime, and model size to measure the model efficiency. Specifically, MACs are measured using PTFLOPS [29] library. The runtime is calculated by the average inference time per frame on the KITTI-15 dataset and the reported value is the average of three runs. Every model is applied to 10 updates during inference on a GTX TITAN V GPU. And the model size is measured in the number of parameters. All models are fine-tuned using the combined dataset from FlyingThings3D, HD1K, KITTI, and Sintel training sets. The evaluation is conducted on the Sintel test set. As illustrated in Figure 4, our proposed SKFlow shows better accuracy compared with GMA with similar MACs ($8.42\%$) and a limited increase in model size ($6.63\%$). The runtime comparison is shown in the supplemental materials. Compared with other architectures that widely adopt the well-optimized

$3 \times 3$ convolution, the runtime of SKFlow is a little bit longer in practice, due to the lack of efficient implementation for large depth-wise kernels in the current PyTorch library.

## 4.3 Occlusion analysis

We split pixels into occluded (Occ) and non-occluded (Noc) with the help of the occlusion maps provided in the Sintel dataset. Following the previous work [16], the model is pre-trained using the C+T schedule and then fine-tuned using the C+T+S+H+K schedule for 150K iterations. The Albedo pass of the Sintel dataset is used to verify the effectiveness of estimating occluded motions and is not used in training. We choose Albedo pass as the test set because it is rendered without illumination effects and thus adheres to the brightness constancy constraint except in the occluded areas, which highlights the performance on occlusions. The results are shown in table 2, where "Noc" denotes non-occluded pixels, "Occ" denotes the occluded pixels and "All" denotes the overall pixels. We also test their performance on the unmatched areas on the Sintel test set. Detailed results are shown in the supplements. Results on the occluded and non-occluded areas in the training and test set demonstrate that our SKFlow method can better resolve the ambiguity caused by occlusion areas, compared with RAFT and GMA models.

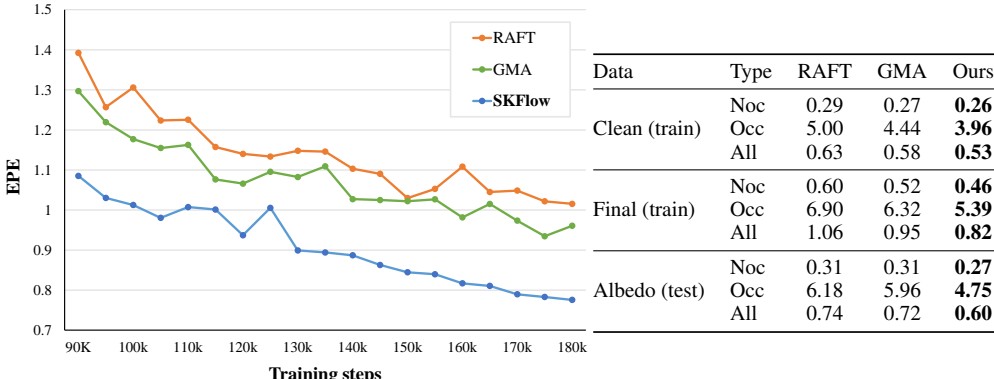

| Data | Type | RAFT | GMA | Ours |
|------|------|------|-----|------|
| Clean (train) | Noc | 0.29 | 0.27 | **0.26** |
| | Occ | 5.00 | 4.44 | **3.96** |
| | All | 0.63 | 0.58 | **0.53** |
| Final (train) | Noc | 0.60 | 0.52 | **0.46** |
| | Occ | 6.90 | 6.32 | **5.39** |
| | All | 1.06 | 0.95 | **0.82** |
| Albedo (test) | Noc | 0.31 | 0.31 | **0.27** |
| | Occ | 6.18 | 5.96 | **4.75** |
| | All | 0.74 | 0.72 | **0.60** |

Figure 5: EPE curve on Sintel during training. All models are trained under the same settings.

Table 2: Performance on occluded and non-occluded areas on the Sintel dataset.

## 4.4 Ablation study

We perform a set of ablation experiments to show the importance of each component in the network design. The results are shown in Figure 4 (b) and Table 5 respectively. All ablated versions are trained using the C+T schedule. The settings which are used in our final model are underlined. In the following, we describe each of the experiments in more detail.

**Training schedules.** In particular, we find that the performance of our proposed SKFlow keeps improving when training steps are increasing. Specifically, we train the model with additional 30K steps on the FlyingThings dataset and additional 60K steps for fine-tuning on the Sintel dataset. For a fair comparison, we then train GMA and RAFT using the same settings. As illustrated in Figure 5, the EPE of SKFlow keeps decreasing while the training steps are increasing, and an obvious improvement in performance can be obtained compared to the RAFT and GMA. We argue that large receptive fields are inherently helpful to capture more motion information and raise the upper bound of the learning ability of networks, although it tends to be hard to optimize.

**Kernel sizes.** We explore the impact of different kernel sizes. Specifically, the kernel size is increased from 1 to 31, growing $2 \times$ each time. As shown in Table 5, we can see that kernel size $15 \times 15$ achieves the best overall performance. Although it obtains $0.89\%$ and $0.92\%$ lower performance than $7 \times 7$ kernel on KITTI, it achieves $6.98\%$ and $5.56\%$ higher in performance on Sintel clean and final pass. The comparison with other designs is shown in Figure4 4 (b). We also observe much lower performance when the kernel size is 31, especially on a small dataset such as

Table 3: Results on the FlyingChairs validation set.

| Model | FlyingChairs | Parameters |
|---|---|---|
| RAFT | 0.85 | 5.26M |
| RAFT-Deep | 0.85 | 6.00M |
| SKFlow-RAFT | 0.83 | 5.59M |
| GMA | 0.80 | 5.88M |
| GMA-Deep | 0.83 | 6.25M |
| SKFlow-GMA (Ours) | **0.78** | 6.27M |

Table 4: Results on the Sintel training set.

| Model | Clean | Final |
|---|---|---|
| GMAMoE + GMAGRU | 1.30 | 2.74 |
| (GMAMoE + GMAGRU)* | 1.36 | 2.72 |
| SKMoE + GMAGRU | 1.28 | 2.56 |
| SKMoE + GMA-LargeGRU | 1.32 | 2.54 |
| SKMoE + SKBlock-Small | 1.25 | 2.56 |
| SKMoE + SKBlock (Ours) | **1.22** | **2.46** |

KITTI, which indicates the optimization of the extremely large kernel is still a challenge given the limited number of samples in optical flow datasets.

**Super kernel block components.** We explore the effect of each component in Section 3.1). Specifically, "Res" denotes the residual connections and "Aux" denotes the Auxiliary small kernel. From Table 5 "Components" we can see that by using both the residual connections and the auxiliary small kernel, the overall performance improves significantly. (comparable result 2.46 vs 2.45 on Sintel Final). Nevertheless, without the auxiliary kernel, there will be a significant drop for a small dataset like KITTI, indicating the importance of the auxiliary kernel on optimization.

**Super kernel block architecture.** We explore the performance of three hybrid architecture designs in Section 3.1, as shown in Table 5 "Architecture". We can see that the conical connection design achieves the best performance, showing a strong capability to optimize optical flow networks with large kernels. It is different from the existing work [7] showing that the parallel re-parameterization structure performs well in high-level tasks such as image classification. On optical flow estimation, our conical structure obtains better performance. There might be two reasons: **(1)** Differences in the distribution and size of datasets. Compared with large datasets such as ImageNet where samples are from real life, most optical flow datasets are synthetic and contain much smaller samples, which makes it harder for large kernels to optimize. **(2)** Different focus of tasks. Compared with optical flow tasks, high-level vision tasks pay more attention to semantic information instead of per-pixel correspondence between two images.

**Auxiliary kernel size.** We also explore different sizes for the auxiliary kernel. According to the results shown in Table 5 "Aux kernel size", the most effective auxiliary kernel size is $1 \times 1$. Although $3 \times 3$ achieves a slightly better performance on Sintel, $1 \times 1$ kernel outperforms $3 \times 3$ kernel on Fl-epe and Fl-all of the KITTI dataset with more improvement and less computation.

**Super kernel modules.** We compare the performance of using super kernels in update block and motion encoder and the results are shown in Table 5 "SK modules". Upd. refers to the update block and MoE. denotes the motion encoder. We can see that by using super kernels in the update block the performance is 2.65 on the Sintel final pass, while by using the super kernels in the motion encoder, the performance is 2.56. When super kernels are applied on both the motion encoder and update block in the SKFlow network, the performance on both Sintel and KITTI datasets is further improved significantly.

**Deep layers vs. Large kernels.** As shown in Table 3, adopting deeper layers brings little improvement compared with using large kernels. We argue that the deeper layers may lead to harder optimization, and then the receptive field will behave like that in shallow networks, as mentioned in the previous work [7]. All models are trained on the FlyingChairs training set with the same setting. We stack deeper layers in RAFT and GMA decoders so that they have model sizes similar to SKFlow. It also ablates the effect of additional parameters in SKFlow.

**Switch from GRU to SKBlock.** The motivation for replacing GRU with SKBlock is that we find directly increasing the kernel size of GRU architecture tends not to bring performance gain. As shown in Table 4, from GMAGRU to GMA-LargeGRU, the improvement on Sintel (final) tends to be saturated and the performance on Sintel (clean) degrades to a small extent. Nevertheless, from SKBlock-Small to SKBlock, the improvement is quite stable. All models are trained using the C $\rightarrow$ T schedule under the same settings. Note that GMAGRU and SKBlock-Small share the same

Table 5: Ablations on super kernel block designs. Models are trained using the C+T schedule.

| Experiment | Method | Sintel(train) | | KITTI-15(train) | | Parameters |
|---|---|---|---|---|---|---|
| | | Clean | Final | Fl-epe | Fl-all | |
| Large kernel size | 31 | 1.27 | 2.65 | 4.73 | 17.40 | 7.08M |
| | 15 | **1.20** | **2.55** | 4.51 | 16.26 | 6.27M |
| | 7 | 1.29 | 2.70 | **4.47** | **16.11** | 6.08M |
| | 3 | 1.30 | 2.77 | 4.83 | 17.31 | 6.04M |
| | 1 | 1.41 | 2.83 | 7.12 | 21.58 | 6.03M |
| Components | No Res | 1.35 | 2.56 | 4.76 | 17.33 | 6.27M |
| | No Aux | 1.25 | **2.45** | 4.53 | 16.55 | 6.27M |
| | No Res & Aux | 1.33 | 2.61 | 5.04 | 17.38 | 6.27M |
| | With Res & Aux | **1.22** | 2.46 | **4.27** | **15.47** | 6.27M |
| Architecture | Parallel | 1.24 | 2.53 | 4.71 | 17.82 | 6.27M |
| | Funnel | **1.19** | 2.54 | 4.55 | 16.97 | 6.27M |
| | Conical | 1.22 | **2.46** | **4.27** | **15.47** | 6.27M |
| Aux kernel size | 3 | **1.21** | **2.44** | 4.33 | 15.56 | 6.29M |
| | 1 | 1.22 | 2.46 | **4.27** | **15.47** | 6.27M |
| SK modules | Upd. | 1.37 | 2.65 | 4.89 | 16.89 | 5.30M |
| | MoE. | 1.28 | 2.56 | 4.60 | 16.86 | 6.67M |
| | Upd.+MoE. | **1.22** | **2.46** | **4.27** | **15.47** | 6.27M |

kernel size, and GMA-LargeGRU obtains the same kernel size as SKBlock. GMAMoE + GMAGRU denotes the result from the original GMA paper and (GMAMoE + GMAGRU)* is our reproduced result for GMA.

## 4.5 Visualiztions

As shown in Figure 6, we visualize the predicted optical flows on the sample image from the Sintel test set. Results are highlighted using red circles in the figure, which demonstrates that our proposed SKFlow predicts a more accurate flow and captures more details compared with other methods.

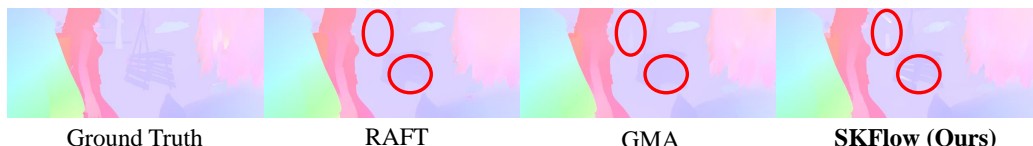

| Ground Truth | RAFT | GMA | **SKFlow (Ours)** |

Figure 6: Visualizations of predicted optical flow among different methods on Sintel test set.

## 5 Conclusions

In this work, we propose SKFlow, which utilizes large receptive fields brought by super kernels to recover the occluded motion in optical flow estimation. To the best of our knowledge, SKFlow is the first to apply the kernel size up to $15 \times 15$ to optical flow networks with significant performance gain and modest computation increase. To leverage the computation cost for the large kernels and attains improved accuracy, SKFlow explores and provides effective and efficient design schemes for applying super kernels, such as conical connections and hybrid depth-wise convolutions. Comprehensive experiments have demonstrated the effectiveness of SKFlow. With less than an $8.42\%$ increase in MACs, SKFlow outperforms the previous state-of-the-art method on the Sintel final pass, ranking 1st among all published methods.

**Acknowledgements.** This work was supported by National Natural Science Foundation of China (No. 62172021).

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
