# OpenReview forum: "SKFlow: Learning Optical Flow with Super Kernels"
_NeurIPS.cc/2022/Conference — NeurIPS 2022 Accept_

### Official Review · Reviewer_dtXq · 2022-06-23

**Rating:** 6
**Confidence:** 4
**Soundness:** 3 good
**Presentation:** 3 good
**Contribution:** 3 good

**Summary:**

This work addresses the problem of optical flow estimation and is based on RAFT and GMA. The paper proposes to use a larger kernel in convolutional neural networks to solve the problem of occlusion occurring in optical flow. A super kernel block with depth-wise and channel-wise convolution is introduced to reduce the computational cost. The experimental result shows state-of-the-art performance.

**Questions:**

1. Line 127 is a little bit unclear about the computation cost. How is the cost obtained in more detail?
2. When is the computational cost of SKFlow smaller than the normal kernel convolutional method? Is the complexity related to the number of input channels and output channels and the size of the kernel? For the example in the paper (kernel size of 15), what is the ratio of cost_ours and cost_normal?
3. In line 276, it is said that the most effective auxiliary kernel size is 1*1. But in that case, isn’t depth-wise convolution just a multiplication? Where does the better performance come from?
4. Experiment: Why is GMA chosen as the baseline? Is there a specific reason? Have other baselines been explored using the super kernel block? If so, what results have been obtained?
5. Has the proposed approach been compared to a traditional large kernel approach? For example, using a kernel size of 15 to replace the super kernel block. Is this possible? How large is the computational cost in comparison to the super kernel block?

Additional questions of interest:

6. The occlusion analysis is only tested on the Albedo pass, how about testing on the Clean pass and the Final pass? How does it perform?

7. Have deeper layers been tried instead of a larger kernel to enlarge the receptive field? Does it perform badly or is it just a reasonable assumption that using a larger kernel will have a better result?

**Limitations:**

The paper did not mention any limitations or negative societal impacts. The latter might be difficult, but optical flow in general has its limitations regarding the estimation of the (projected) motion. Isn't it time to also consider learning to estimate 3D motion or the projected motion instead of optical flow?

**Strengths And Weaknesses:**

Strengths:
1. The paper is clearly written with the baseline method and the contribution.
2. The proposed super kernel block is simple in structure and makes generalization to other networks possible.
3. The ablation study is quite comprehensive.
Weaknesses:
1. Using a larger kernel in optical flow has been proposed before, see line 40 and [7,11,24-26]. The extension to the proposed super kernel is rather incremental.
2. The proposed extension is only applied to the GMA architecture and it remains unclear whether the super kernel block will improve results for other architectures.

---

> ### Author Response · Authors · 2022-08-02
> **Response to reviewer dtXq I**
>
> Thank you for your review and questions. Below we will clarify each question in detail.
>
> **Q1: Using a larger kernel in optical flow has been proposed before, see line 40 and [7, 11, 24-26].**
>
> Thank you for the comment. Nevertheless, these works [7, 11, 24-26] do not apply large kernels in optical flow. Instead, they focus on high-level tasks such as classification and detection, etc. To the best of our knowledge, we are the first to apply super kernels to optical flow estimation and successfully alleviate the classical occlusion issue with super kernels.
>
> As stated in line 40 and the related work section, these works [7, 11, 24-26] inspire us to obtain a larger receptive field by enlarging the kernel size. Nevertheless, it is hard to directly apply them to the optical flow task, as shown in lines 74-85. Therefore, to deal with the challenges of introducing large kernels to optical flow, we propose SKFlow. Compared with these previous works, the problem we aim to address and the proposed solution are quite different.
>
> **Q2: Computation cost in more detail.**
>
> The use of the hyper-parameter $D$ in line 127 indeed makes the formula unclear. Therefore we will remove the parameter $D$ in the clarification below. We are sorry for the ambiguous expression and will revise it in our final version. The total computation cost is obtained as follows. Given the input feature map with the shape of $N\times C_{in} \times H \times W$, the large kernel with size $L \times L$ and the auxiliary kernel with size $S \times S$,  the computation cost of a super kernel block contains two parts:
>
> **(1) Basic components.** As shown in Figure 2, it is the combination of the hybrid depth-wise convolution and point-wise convolution. For the hybrid depth-wise convolution, the computation cost is
> $
> N \times C_{in} \times H \times W \times L^2 + N \times C_{in} \times H \times W \times S^2,
> $
> For the point-wise convolution, the computation cost is
> $
> N \times C_{in} \times 1 \times 1 \times H \times W \times C_{in},
> $
> Then the overall computation for the basic components is
> $
> N \times C_{in} \times H \times W \times (L^2 + S^2 + C_{in}),
> $
>
>
> **(2) Dimension lifters.** As mentioned in line 127, to match the dimension and increase more non-linear transforms, two pairs of $1 \times 1$ convolutions are performed in each super kernel block. Each pair of convolution layers will first transform the input feature map from dimension $C_{in}$ to $\alpha C_{in}$, and then from $\alpha C_{in}$ to $C'$. Namely, the computation is
> $
> N \times H \times W \times C_{in} \times \alpha C_{in} + N \times H \times W \times \alpha C_{in} \times C',
> $
> where $C' = C_{in}$ for the first pair of convolution layers and $C' = C_{out}$ for the second pair. Therefore, the total computation for the dimension lifter is
> $
> N \times H \times W \times \alpha C_{in} \times (C_{in} + C_{out} + C_{in} + C_{in}),
> $
> In summary, the total computation of an SK block is
> $
> N \times H \times W \times C_{in} \times [L^2 + S^2 + C_{in} + \alpha (3C_{in} + C_{out})].
> $
>
>
> **Q3: Factors that affect the SKFlow computation cost and under which circumstances will SKFlow be more efficient.**
>
> The cost ratio is then
> $$
> \frac{Cost_{ours}}{Cost_{normal}} = \frac{N \times H \times W \times C_{in} \times [L^2 + S^2 + C_{in} + \alpha (3C_{in} + C_{out})]}{N \times H \times W \times C_{in} \times C_{out} \times {L^2}} = \frac{1}{C_{out}} + \frac{1}{C_{out}}\frac{S^2}{L^2} + \frac{1+3\alpha}{L^2}\frac{C_{in}}{C_{out}} + \frac{\alpha}{L^2},
> $$
> We can learn that the ratio is related to the number of the input channel, output channel, and kernel size.
>
> For the example in the paper, $\alpha = 1.5$, $S=1$, $L=15$. $C_{in}$ is usually equal to or twice as much as $C_{out}$, here we set $C_{in} = 256$ and $C_{out} = 128$. Then $\frac{Cost_{ours}}{Cost_{normal}} \approx 0.063$. We could learn that the computation of normal large convolution is quite expensive and the optimization of SKFlow is necessary.
>
> Compared with normal large convolution layers, SKFlow is more efficient when $L$ is large enough and $C_{in}$ is not too greater than $C_{out}$. Let $\frac{Cost_{ours}}{Cost_{normal}} < 1$, then we could derive that  $L^2 > \frac{S^2}{C_{out}-1} + \frac{C_{in}(1+3\alpha)}{C_{out}-1} + \frac{\alpha C_{out}}{C_{out}-1}$. In practice, $C_{out} \gg S^2 \ge 1$. Approximately,
> $
> L^2 > \frac{(1+3\alpha)C_{in}}{C_{out}} + \alpha,
> $
> which holds for many cases. For instance, let $L=15$ and $\alpha = 1.5$. The inequality holds **unless** $C_{in}$ is nearly 40 times larger than $C_{out}$, which is quite rare in optical flow networks. In that case, SKFlow becomes less efficient because small $C_{out}$ leads to a limited reduction of computation, and the great $C_{in}$ makes the computation of dimension lifters not negligible.

---

> > ### Author Response · Authors · 2022-08-02
> > **Response to reviewer dtXq II**
> >
> > **Q4: Why does 1 x 1 auxiliary kernel perform better than 3x3?**
> >
> > Actually, the performance of the 1x1 and 3x3 auxiliary kernels is quite close. 3x3 auxiliary kernel even performs better on Sintel. We finally select the 1x1 kernel due to its less computation and slightly better overall performance. As for the effectiveness, to some extent, we argue that the $1 \times 1$ depthwise auxiliary kernel behaves like a kind of channel attention mechanism. In this case, it highlights motions and objects captured in different channels.
> >
> > **Q5:  Why GMA is the baseline and if SKFlow could improve other methods?**
> >
> > We choose GMA as our main baseline because it achieves state-of-the-art performance on the Sintel benchmark till the paper is written. Notably, based on such an effective method, our proposed SKFlow could still achieve an obvious improvement on standard benchmarks.
> >
> > Besides, we further explore the effectiveness of super kernel block applied to other methods, e.g., RAFT. As shown in the following table, models are tested on the challenging Sintel benchmark. All models are trained using the FlyingChairs $\rightarrow$  FlyingThings $\rightarrow$ Sintel schedule. Based on RAFT, the SKFlow method still achieves an obvious improvement both on the training and test sets. Notably, as mentioned in the response to Q7, the improvement is not only in the non-occluded areas but also in the occluded areas.
> >
> > | Model     | clean (train) | final (train) | clean (test) | final (test) | Parameters |
> > | --------- | :-----------: | :-----------: | :----------: | :----------: | :--------: |
> > | RAFT      |     0.76      |     1.22      |     1.61     |     2.86     |   5.26M    |
> > | Ours-RAFT |     0.62      |     0.91      |     1.46     |     2.61     |   5.59M    |
> > | GMA       |     0.62      |     1.06      |     1.39     |     2.47     |   5.88M    |
> > | Ours      |   **0.52**    |   **0.78**    |   **1.28**   |   **2.23**   |   6.27M    |
> >
> > **Q6: If it is possible to directly adopt the large convolution and how large the computation cost is.**
> >
> > Unluckily, as shown in the response to Q2, directly applying a $15 \times 15$ normal convolution layer is a little bit unpractical due to the dozens of times the computation cost. Therefore the reduction of computation is necessary.
> >
> > **Q7: How about testing the occlusion analysis on the Clean and Final test set and how does it perform?**
> >
> > Thank you for your question. We can also test the occluded areas on the Clean and Final pass with statistics from the official Sintel leaderboard. We could not get the accurate occluded areas since we authors have no access to occlusion maps on the test set. But the Sintel server provides the results on the **matched and unmatched** areas, which are the extension of the non-occluded and occluded areas. According to the Sintel website, the matched areas are regions that remain visible in adjacent frames, and the unmatched denote regions visible only in one of two adjacent frames. Therefore, we could still evaluate the performance in the generalized occluded areas. The results are as follows:
> >
> > | Model     | Matched (clean) | Unmatched (clean) | Matched (final) | Unmatched (final) |
> > | --------- | :-------------: | :---------------: | :-------------: | :---------------: |
> > | RAFT      |      0.623      |       9.647       |      1.405      |      14.680       |
> > | Ours-RAFT |      0.617      |       8.346       |      1.288      |      13.352       |
> > | GMA       |      0.582      |       7.963       |      1.241      |      12.501       |
> > | Ours      |    **0.554**    |     **7.239**     |    **1.145**    |    **11.511**     |
> >
> > **Q8: How about adopting deeper layers to enlarge the receptive field?**
> >
> > We have tried stacking deeper layers to enlarge the receptive field but there is little improvement, as shown in the following table. Given limited rebuttal time, all models are trained on the FlyingChairs training set with the same setting. We argue that the deeper layers may lead to harder optimization, and then the receptive field will behave like that in shallow networks, as mentioned in the previous work [1].
> >
> > | Model     | FlyingChairs (validation set) | Parameters |
> > | --------- | :---------------------------: | :--------: |
> > | RAFT      |             0.85              |   5.26M    |
> > | RAFT-Deep |             0.85              |   6.00M    |
> > | Ours-RAFT |             0.83              |   5.59M    |
> > | GMA       |             0.80              |   5.88M    |
> > | GMA-Deep  |             0.83              |   6.25M    |
> > | Ours      |           **0.78**            |   6.27M    |
> >
> > [1] Ding X, Zhang X, Han J, et al. Scaling up your kernels to 31x31: Revisiting large kernel design in cnns[C]//Proceedings of the IEEE/CVF Conference on Computer Vision and Pattern Recognition. 2022: 11963-11975.

---

> > > ### Author Response · Authors · 2022-08-02
> > > **Response to reviewer dtXq III**
> > >
> > > **Q9: Limitations of the proposed method.**
> > >
> > > - **Time efficiency.** Currently, most libraries are lacking in efficient optimization for the computation of depthwise convolution with large kernels. Therefore, compared with the well-optimized normal $3 \times 3$ convolution, the super kernel block may be a little less efficient on devices like GPU now.
> > > - **3D motion estimation.**  3D contextual information could be helpful for addressing the occlusion issue. In this paper, we mainly focus on the estimation of optical flow. We would explore the application of SKBlock in 3D motion estimation in future work.

---

> > > > ### Comment · Reviewer_dtXq · 2022-08-09
> > > > **Reflection on rebuttal**
> > > >
> > > > The authors answered most of my questions well. Most importantly they show that the computational cost is only a fraction of the normal convolutional layers with large kernel, and the proposed super kernel works well on both RAFT and GMA. But given the fact that large kernel has been explored in other high-level tasks, and the super kernel is only a small modification of the existing model, I only raise my assessment slightly.

---

### Official Review · Reviewer_64b1 · 2022-07-09

**Rating:** 9
**Confidence:** 5
**Soundness:** 4 excellent
**Presentation:** 4 excellent
**Contribution:** 4 excellent

**Summary:**

In this work, the authors proposed a simple but effective way to handle occlusion in optical flow estimation. The main idea is to use a kernel with a large reception field. Unlike previous work that simply increased the depth of network to increase network, the authors proposed a new super-kernel block, which uses a super block that consists of a large and a small kernel. Experiments have demonstrated the proposed network can significantly boost the accuracy of optical flow, with minimal additional compute.

**Questions:**

### I only have two minor questions about this work.

Another way to increase the receptive field without introducing too many network parameters is dilated convolution. It would be great if the authors can discuss Pros and Cons of proposed approach compared to dilated convolution.

For efficiency evaluation, I think it is also important to report actual runtime, in addition to FLOPS/MACs, as due to cache locality and other memory issues, FLOPS/MACs do not always lead to a low execution time, particularly when a large kernel is used like this work. For completeness, it is better to report runtime on a certain device.

### I also have a few suggestions

The authors did not need to reply to this question in rebuttal, but would be great to answer them in the camera-ready version.

Funnel v.s. Conical. In theory, convolutional kernels are mostly interchangeable, so I would expect funnel v.s. Conical should achieve similar results. Experiments actually show that conical performs slightly better than funnel (Table 3). It would be great if authors could share some insights, even preliminarily, why funnel performs better, and why it performs better than parallel on the selected datasets (I understood datasets could make difference, but would still be interesting to the audience if the authors give more insights).

The proposed super kernel is a very general design. It would be great if the authors could also discuss the potential usage of this structure in other image processing tasks in conclusions / future work.


**Limitations:**

No as far as I know.

**Strengths And Weaknesses:**

This is a very high quality work. The authors proposed a very simple but also very effective network to increase the receptive field of the flow estimation network. The authors achieved the best accuracy in the Sintel benchmark, and also did a very thorough experiment to demonstrate effectiveness of the proposed network on different setups and how different alternative designs may work. At last, although this work mainly focuses on motion estimation, the proposed architecture is very general, and I believe it could benefit many other image processing tasks and even beyond.

I don't find any particular weak point in this work.

---

> ### Author Response · Authors · 2022-08-02
> **Response to reviewer 64b1**
>
> Thank you for your review and the constructive advice on our work. These useful suggestions will surely help us to further improve our work. We would dig deeper into these questions and try to figure them out in later versions.
>
> **Q1: Pros and Cons of the proposed approach compared to dilated convolution.**
>
> Thank you for your comment. Dilated convolution is indeed an efficient way to obtain large receptive fields and we then implement a dilated version of our proposed method. In our dilated version, the large $15 \times 15$ depth-wise convolutions are replaced with a $9 \times 9$ dilated convolutions with a dilation rate of 2. Quantitive results are shown in the following table. Models are trained using the FlyingChairs $\rightarrow$ FlyingThings schedule and then validated on Sintel. GMA denotes the result in the original paper and GMA* denotes our reproduced result.
>
> | Model        | Sintel (clean) | Sintel (final) | Parameters |
> | ------------ | :------------: | :------------: | :--------: |
> | GMA          |      1.30      |      2.74      |   5.88M    |
> | GMA*         |      1.36      |      2.72      |   5.88M    |
> | Ours-Dilated |      1.32      |      2.55      |   6.10M    |
> | Ours         |    **1.22**    |    **2.46**    |   6.27M    |
>
> **Cons of Our method:** We could learn that the dilated convolution is indeed more efficient, as shown in the table. With a similar receptive field, the dilated version costs less computation and achieves a better overall performance than the baseline GMA.
>
> **Pros of Our method:** Our proposed method obtains a strong cross-dataset generalization ability. With a modest increase in computation, our method achieves the leading performance on both the clean and final pass.
>
> **Discussion:** Although our method and the dilated version have a receptive field of a similar size, there is a small gap in the performance. We argue that the gap may be caused by the gridding effect [1]. Namely, the receptive field of a dilated convolution kernel covers an area with checkerboard patterns. Therefore, the sampled locations contribute to the calculation but the neighboring information is lost. In this case, the gridding effect leads to two issues that might affect the estimating of per-pixel motion: (1) Absence of local information. (2) Irrelevant information across large distances due to the sparse sample of input.
>
> Nevertheless, given the efficiency and the various applications of dilated convolution, we believe that how to more properly apply it to optical flow networks is still a question worth further studying. We hope there emerge more comprehensive and deeper explorations in future works.
>
> **Q2: Comparison of runtime.**
>
> Thank you for your suggestions. We test all models on a GTX TITAN V GPU using the PyTorch library. The runtime is calculated by the average inference time per frame on the KITTI-15 dataset and the reported value is the average of three runs. Every model is applied to 10 iterations during inference. Please note that, due to the lack of efficient implementation for large depth-wise and dilated kernels in the current PyTorch library, the latency would be a little bit longer in practice, compared with other architectures that widely adopt the well-optimized $3 \times 3$ convolution.
>
> | Model        | Time  |
> | ------------ | :---: |
> | RAFT         | 0.13s |
> | GMA          | 0.16s |
> | Ours-Dilated | 0.21s |
> | Ours         | 0.22s |
>
>
>
> [1] Wang P, Chen P, Yuan Y, et al. Understanding convolution for semantic segmentation[C]//2018 IEEE winter conference on applications of computer vision (WACV). Ieee, 2018: 1451-1460.

---

> > ### Comment · Reviewer_64b1 · 2022-08-08
> > **Feedbacks**
> >
> > Thanks for sharing this additional experiment dilated convolution and runtime. It is very clear that the proposed algorithms perform better than dilated conv and would be nice include this discussion to the main paper or supplementary material.
> >
> > As stated in the first round, this is a very high quality work with a simple and effective idea. I will keep my rating as strong acceptance.

---

> > > ### Author Response · Authors · 2022-08-09
> > > **Response to feedbacks of reviewer 64b1**
> > >
> > > Thank you for your feedback. We will include this discussion in the supplementary material.

---

### Official Review · Reviewer_Qxdd · 2022-07-10

**Rating:** 4
**Confidence:** 5
**Soundness:** 2 fair
**Presentation:** 2 fair
**Contribution:** 2 fair

**Summary:**

This paper proposes a CNN-based method that deals with optical flow estimation, especially to tackle occluded areas. To that end, the proposed method studies the relationship between the receptive fields of the network and the effectiveness of recovering occluded motions. The proposed method designs the network using super kernel blocks, which are composed of small size depth-wise convolutions followed by a set of large-size convolutions. Comprehensive experiments demonstrate that the proposed method is quite effective and outperforms the state-of-the-art baseline methods.

**Questions:**

1. Usually KITTI-2015 dataset contains more large displacement / fast motion (than Sintel Dataset), which introduces considerably more occluded areas. But the performance of the network is not better than its predecessor network RAFT. Can the authors provide more discussion about it?

**Limitations:**

The main limitation is that the paper does not describe clearly the reason why the Conical architecture is the key reason to improve the performance of the proposed network in dealing with occlusion areas. It is not straightforward/intuitive relationship between the receptive field and the occluded area. So it requires the paper to make a more sound analysis and insightful theoretical development.

**Strengths And Weaknesses:**

Strengths
1. The proposed method consistently outperforms the baseline methods on mainstream optical flow benchmarking datasets, including Sintel (synthetic), KITTI-15(real).  The proposed method is submitted to the Sintel benchmark platform and obtained the state-of-the-art ranking on that dataset.

2. This paper is generally clearly written with a proper literature review. The introduction part has stated the motivation of the paper clearly and specifies the challenges of the existing methods as well as the key component to deal with the challenges.  The method section has delivered a clear architecture by using visual graphs and proper citation. The experiments part have gone through a systematic way to run benchmarks.

Weaknesses
1. This paper's main problem lies in the theory part of the approach section. It does not specify the theory behind the super kernel to validate the reason for the effectiveness. Even though, we understand that the architecture outperforms other methods in the experiments. But it lacks the reasoning why the architecture is designed in that way in the first place.

2. In the experiment section, the description says the models are trained on more challenging data (i.e. Sintel clean and final) but tested on the intrinsic components of the scene (i.e. Albedo), which usually cannot validate the effectiveness of the methods. But in Table 1, the results are conducted on the clean and final datasets.

3. The authors are suggested to conduct one more ablation study to verify that the improved performance is not coming from more parameters compared with the baseline RAFT and GMA method by running a vanilla network with the same number of parameter networks.

4. Based on the ablation study, it seems that SK modules ( motion encoder and updater ) affect the performance much more than the conical architecture compared with other architecture settings. But the paper claims that the key effective components should be the conical design of the network. Can the authors explain and validate more on this issue?




5. Many typos:
i). All the equations (1) -(11) lack punctuation.
ii). Line 173, 182, etc, the short headline should not come with full stop.
iii). e

---

> ### Author Response · Authors · 2022-08-02
> **Response to reviewer Qxdd**
>
> Thank you for your review and suggestions. Below are our clarifications for specific questions.
>
> **Q1: The reason why the architecture is designed in that way.**
>
> Thank you for your comment. The introduction of super kernels is to enlarge receptive fields to resolve the ambiguity of occluded areas. And the motivation behind the overall super kernel architecture design mainly contains the following three points:
>
> 1. Firstly, directly applying normal large convolution layers is too expensive and the reduction of cost is quite necessary. In this case, the separable convolution is introduced.
> 2. Given the multi-scale objects and motions between frames, we additionally introduce the auxiliary small kernel to help capture the small patterns. Notably, the vanilla architecture without the auxiliary kernel still outperforms the baseline model, as shown in Table 3. And adding the auxiliary kernel will further improve the performance, whether in a parallel, funnel, or conical way. This strongly supports our idea to adopt the auxiliary kernel.
> 3.  To alleviate the optimization issue, we widely adopt residual connections. From the ablation section, we can learn that the introduction of residual connections is quite effective.
>
> Following the principles above, we further design three blocks (Parallel, Conical, and Funnel). Notably, all these methods achieve obvious improvements compared with baseline GMA. Among all three architectures, we finally select the conical block due to its relatively better performance.
>
> **Q2: Descriptions on the test set.**
>
> We are sorry for the unclear description. We test the results not only on the challenging Sintel clean and final test set (Table 1) but also on the Albedo set (Table 2). The test on Albedo set is mainly to highlight the analysis of occluded areas, as mentioned in line 228. We will revise the expression in the final version and hope the edited section clarifies the ambiguity.
>
> **Q3: Ablations on additional parameters.**
>
> Thank you for your detailed suggestions. Considering that RAFT, GMA, and ours adopt the same encoder, we deepen the decoder of RAFT and GMA to make sure they obtain the similar number of parameters to our methods. Given the limited rebuttal time, all models are trained on the FlyingChairs training set with the same settings and validated on the validation set.
>
> As shown in the table, our proposed method still obtains improvements with 6.63%+ and 0.32%+ parameters compared to GMA and GMA-Deep. Ours-RAFT is the method that applies SKBlock to RAFT. Compared with RAFT and RAFT-Deep, Ours-RAFT achieves higher performance with 6.27%+ and 7.33%- parameters. Notably, the additional parameters of RAFT-Deep and GMA-Deep do not further improve the performance. Although deeper layers lead to larger receptive fields, we contend that it also bring issues for optimization.
>
> | Model     | FlyingChairs (validation set) | Parameters |
> | --------- | :---------------------------: | :--------: |
> | RAFT      |             0.85              |   5.26M    |
> | RAFT-Deep |             0.85              |   6.00M    |
> | Ours-RAFT |             0.83              |   5.59M    |
> | GMA       |             0.80              |   5.88M    |
> | GMA-Deep  |             0.83              |   6.25M    |
> | Ours      |           **0.78**            |   6.27M    |
>
> **Q4:  The paper does not describe clearly the reason why Conical architecture is the key reason for improving the performance.**
>
> Thank you for your comment. Actually, we believe that the key component is the overall design scheme for super kernel networks instead of ways to aggregate auxiliary kernels. We do not think conical design is the key effective component. In fact, from Table 3, we could learn that the performance gap between the conical design and the other architectures (especially, the funnel design) is not very large and all three architectures achieve an obvious improvement on the baseline model. We will emphasize this point in the revised version and hope it could clarify the question.
>
> **Q5: Typos.**
>
> Thank you very much for your detailed comments. We will correct all typos and double-check the writing in the final version.
>
> **Q6: Comparison with the predecessor RAFT on KITTI-15.**
>
> KITTI-15 is indeed quite challenging. Nevertheless, our method outperforms RAFT both on the training and test set, as shown in Table 1. The optimization on KITTI is harder compared with other datasets. In addition to the challenging scenes in KITTI, we argue that the limited training set (200 samples) also makes the optimization for large kernel networks harder.

---

### Official Review · Reviewer_vu4D · 2022-07-10

**Rating:** 6
**Confidence:** 3
**Soundness:** 3 good
**Presentation:** 3 good
**Contribution:** 2 fair

**Summary:**

The authors propose to improve optical flow estimation performance in occluded regions by increasing the receptive field size of the network. The idea itself is straightforward, but there are technical challenges in implementing it. In particular, optical flow datasets are small in size, thus simply optimizing a very deep network is not a viable option. Directly increasing the kernel size also does not work well due to especially high memory requirements during flow estimation and optimization difficulties associated with large kernel networks. To address these issues, the authors propose two architectural tricks - ones is using depth-wise-separable convolutions, and another is adding another 1x1 convolutional layer before the large kernel convolution to simplify optimization. Extensive experimental analysis demonstrates that the resulting approach outperforms the state-of-the-art by non-negligible margins with particularly large improvements in occluded regions. An extensive ablation analysis demonstrates the importance of the most of the architectural decisions for the final performance.

**Questions:**

Please provide a motivation to for the switch from GRU to a residual network and ablate it.

Please report the number of iterations used during inference and explain the choice of lambda.

**Strengths And Weaknesses:**

Strengths:

The proposed idea of increasing the performance of optical flow estimation in occluded regions by increasing the size of the receptive field is sound.

The proposed architectural tweaks to the GMA architecture to make training with large kernels viable are also sound and their effectiveness is experimentally demonstrated.

The final approach outperforms the state-of-the-art in most scenarios, with improvements being especially significant in occluded regions.

An extensive abalone analysis is provided.

The paper is readable.


Weakness:

The novelty is mainly in the architectural tweaks to the GMA architecture to make training it with large kernels on small datasets viable.

There are still quite a lot of grammatical mistakes in the text.

Although the ablation analysis is quite extensive, a few important points are not addressed. In particular, the switch from a GRU to a residual network is neither motivated nor ablated, and the choice of the hyper parameters is not explained (number of iterations during inference, lambda parameter in the loss).

---

> ### Author Response · Authors · 2022-08-02
> **Response to reviewer vu4D**
>
> Thank you for your review and the time you put into it. Below are our responses to specific questions.
>
> **Q1: The motivation and ablation for the switch from GRU to residual networks.**
>
> Thank you for your comment. The motivation is that we find directly increasing the kernel size of GRU architecture tends not to bring performance gain. As shown in the following table, from GMAGRU to GMAGRU-LargeKernel, the improvement on Sintel (final) tends to be saturated and the performance on Sintel (clean) degrades to a small extent. Nevertheless, from SKBlock-SmallKernel to SKBlock, the improvement is quite stable. Please note that GMAGRU and SKBlock-SmallKernel share the same kernel size, and GMAGRU-LargeKernel obtains the same kernel size as SKBlock. All models are trained using the FlyingChairs $\rightarrow$ FlyingThings schedule under the same settings. GMAMoE + GMAGRU denotes the result from the original GMA paper [1] and (GMAMoE + GMAGRU)* is our reproduced result for GMA.
>
> | Model                       | Sintel (clean) | Sintel (final) |
> | --------------------------- | :------------: | :------------: |
> | GMAMoE + GMAGRU             |      1.30      |      2.74      |
> | (GMAMoE + GMAGRU)*          |      1.36      |      2.72      |
> | SKMoE + GMAGRU              |      1.28      |      2.56      |
> | SKMoE + GMAGRU-LargeKernel  |      1.32      |      2.54      |
> | SKMoE + SKBlock-SmallKernel |      1.25      |      2.56      |
> | SKMoE + SKBlock (Ours)      |    **1.22**    |    **2.46**    |
>
> **Q2: The number of iterations during inference and the choice of lambda.**
>
> **The number of iterations.** We apply 15 iterations when inferring on Sintel and 12 iterations on KITTI. 15 and 12 are selected given the tradeoff between efficiency and accuracy. The accuracy tends to converge after these iterations. Notably, our proposed SKFlow takes fewer iterations to converge, compared with GMA[1] and RAFT [2] which apply 32 iterations on Sintel and 24 iterations on KITTI.
>
> **The choice of lambda.** Its value is usually empirically determined. For a fair comparison, we set it to 0.8, the same as GMA[1] and RAFT[2].
>
> **Q3: The novelty is mainly in the architectural tweaks to the GMA architecture.**
>
> Rather than tweaks to the GMA architecture, we mainly focus on how to successfully introduce kernels with super sizes to address the occlusion issue in optical flow estimation. Then we propose effective design schemes for super kernels and explore three architecture designs for the super kernel block. Actually, the proposed SKBlock is quite flexible and not only works for the GMA architecture. The following table indicates that it could also achieve improvements on RAFT. In this paper, we choose GMA as the main baseline because it achieves state-of-the-art performance on the challenging Sintel benchmark till the paper is written. But SKBlock could be easily adopted by other flow networks.
>
> | Model     | clean (train) | final (train) | clean (test) | final (test) | Parameters |
> | --------- | :-----------: | :-----------: | :----------: | :----------: | :--------: |
> | RAFT      |     0.76      |     1.22      |     1.61     |     2.86     |   5.26M    |
> | Ours-RAFT |     0.62      |     0.91      |     1.46     |     2.61     |   5.59M    |
> | GMA       |     0.62      |     1.06      |     1.39     |     2.47     |   5.88M    |
> | Ours      |   **0.52**    |   **0.78**    |   **1.28**   |   **2.23**   |   6.27M    |
>
> Notably, as shown in the following table, the improvement occurs not only in the matched areas (regions that remain visible in adjacent frames) but also in the unmatched areas (regions visible only in one of two adjacent frames) on the test set, which further demonstrates our effectiveness.
>
> | Model     | Matched (clean) | Unmatched (clean) | Matched (final) | Unmatched (final) |
> | --------- | :-------------: | :---------------: | :-------------: | :---------------: |
> | RAFT      |      0.623      |       9.647       |      1.405      |      14.680       |
> | Ours-RAFT |      0.617      |       8.346       |      1.288      |      13.352       |
> | GMA       |      0.582      |       7.963       |      1.241      |      12.501       |
> | Ours      |    **0.554**    |     **7.239**     |    **1.145**    |    **11.511**     |
>
>
>
>
>
> **Q4: Grammar mistakes in the text.**
>
> Thank you for your detailed review. We will double check the writing and revise all typos in the final version.
>
>
>
> [1] Shihao Jiang, Dylan Campbell, Yao Lu, Hongdong Li, and Richard Hartley. Learning to estimate hidden motions with global motion aggregation. In Proceedings of the IEEE/CVF International Conference on Computer Vision, pages 9772–9781, 2021.
>
> [2]  Zachary Teed and Jia Deng. Raft: Recurrent all-pairs field transforms for optical flow. In European conference on computer vision, pages 402–419. Springer, 2020.

---

> > ### Comment · Reviewer_vu4D · 2022-08-08
> > **Re:re**
> >
> > I thank the authors for addressing my comments. One question which remains unaddressed is the intuition behind the lack of performance gains for the GRU. Do you think it is a fundamental issue with your approach?

---

> > > ### Author Response · Authors · 2022-08-09
> > > **Re:re:re**
> > >
> > > We thank the reviewer for the response. We do not regard it as a fundamental issue. Firstly, we contend that the relatively poor performance of ```SKMoE + GMAGRU-LargeKernel``` makes sense because it directly increases the kernel size and unavoidably encounters the optimization issue. This issue might become more conspicuous in GRU. As gated operators with large kernels are harder to optimize, less accurate gated signals would be provided, and more mistakes would be made in early refinements. Besides, our proposed design scheme aims to explore ways of applying large kernels to convolution layers to handle the occlusion issue. The comparison between convolution layers and GRU is not the focus of this work.

---

### Meta-Review · Area_Chair_TopQ · 2022-08-21

**Recommendation:** Accept
**Confidence:** Certain

**Metareview:**

The paper proposes an optical flow estimation network using very large convolution kernels. Three reviewers consider the paper above the bar, while one considers it below the bar. After consulting the paper, reviews, and rebuttal, the area chair decides to side with the positive reviewers due to the strong optical flow estimation results in the benchmark datasets. The AC believes the paper could provide the field some inspiration to look into large convolution kernels for other visual recognition and processing task.

**Award:**

No

---

### Decision · Program_Chairs · 2022-09-14

Accept